# Transcatheter Interventions for Neonates with Congenital Heart Disease: A Review

**DOI:** 10.3390/diagnostics13162673

**Published:** 2023-08-14

**Authors:** Giovanni Meliota, Ugo Vairo

**Affiliations:** Pediatric Cardiology, Giovanni XXIII Pediatric Hospital, 70126 Bari, Italy; ugo.vairo@policlinico.ba.it

**Keywords:** transcatheter interventions, premature infants, patent ductus arteriosus, aortic valve stenosis, aortic coarctation, PDA stent, RVOT stent

## Abstract

Newborns with congenital heart disease often require interventions linked to high morbidity and mortality rates. In the last few decades, many transcatheter interventions have become the first-line treatments for some critical conditions in the neonatal period. A catheter-based approach provides several advantages in terms of procedural time, length of hospitalization, repeatability and neurodevelopmental issues (usually related to cardiopulmonary bypass). The main transcatheter procedures will be reviewed, as they are now valid alternatives to conventional surgical management.

## 1. Introduction

Newborns with congenital heart disease (CHD) frequently require early interventions after birth. Surgical or transcatheter interventions are mandatory in critical CHD and are linked to high rates of morbidity and mortality [1,2]. With the goal of improving outcomes, enormous advancements in percutaneous interventions over the last three to four decades have increased the preference for a transcatheter or hybrid strategy where possible.

Transcatheter interventions provide a number of benefits which are particularly important in critically ill patients. A catheter-based approach is less invasive, repeatable and offers potential anatomical benefits. It avoids cardiopulmonary bypass and therefore reduces neurodevelopmental issues. In the present work, we will review the most common transcatheter interventions performed in the neonatal period to highlight the current state of the art and areas of potential advancement.

## 2. Atrial Septal Communication

The presence of an unrestrictive atrial communication is crucial for survival during the neonatal period in some forms of CHD. In neonates with transposition of great arteries, inadequate blood mixing due to a restrictive atrial communication causes severe hypoxemia. Thus, a balloon atrial septostomy (BAS) is often required [3]. For the same reason, neonates with double-outlet right ventricle and transposition physiology require BAS. Adequate atrial communication is also important to offload pressure and increase cardiac output in either left or right heart-obstructive CHD. Therefore, BAS may be indicated in tricuspid atresia and pulmonary atresia with intact ventricular septum (Table 1). Atrial communication is enlarged by pulling back a non-compliant balloon, which creates a tear in the thin tissue of the fossa ovalis. BAS is currently an effective procedure with a low rate of serious complications. When a more durable communication is needed or in a case of an abnormally thick atrial septum, cutting balloons may be considered in association with static balloon dilation [4,5].

Newborns with hypoplastic left heart syndrome (HLHS) and restrictive atrial communication experience severe hemodynamic instability due to left atrium egress obstruction, leading to high pulmonary venous pressure and pulmonary edema. In these patients, the traditional BAS is often ineffective, because the atrial septum is usually abnormally thick and leftward displaced. Perforation of the septum may be performed with radio-frequency, and this may be followed by static balloon septoplasty and/or stent implantation (Figure 1) [4]. Techniques are described to create a diabolo configuration of the stent in order to achieve stent stability and minimize the risk of embolization [6]. In a single-center retrospective study by Gossett et al., HLHS patients who received a catheter-based decompression of the left atrium showed a better survival rate compared to the historical cohort directly undergoing a Norwood procedure [7].

## 3. Ductus-Dependent Pulmonary Blood Flow CHD

Neonates with ductal-dependent pulmonary blood flow (PBF) require neonatal repair or palliation with a stable source of PBF until the next stage of surgery. The modified Blalock–Taussig shunt (mBTS) has been used for decades as the first choice of palliation, but it has significant morbidity and mortality incidence (7% and 13%), which has not declined over time [8,9,10].

Gibbs et al. [11] published the first report on implantation of a patent ductus arteriosus (PDA) stent in 1992, with the aim of preventing the non-negligible mortality incidence linked to surgical shunts. The use of rigid bare stents and an inadequate coverage of the entire length of the ductus arteriosus, which usually caused duct constriction and clinical problems, contributed to the poor first outcomes. Since then, ductal stenting has become a trustworthy alternative to the mBTS thanks to a number of technical advancements, such as the use of pre-mounted coronary stents [12]. According to multiple recent research efforts comparing outcomes for mBTS and PDA stenting, PDA stenting shows comparable or improved mortality rates, shorter stays in intensive care units and hospitals, lower rates of complications, lower rates use of postprocedural extracorporeal membrane oxygenation [13,14,15,16] and more symmetric growth of pulmonary arteries [17]. Patients undergoing PDA stent implantation experience higher reintervention rates than do patients who receive mBTS. In some instances, it is a planned reintervention to improve final outcome, such as a repeat stent dilation, enabling additional somatic and pulmonary artery growth [18].

Appropriate pre-procedural imaging should delineate key PDA features for interventional planning: ductal origin, morphology (e.g., straight, tortuous) [19] and ductal length. A comprehensive transthoracic echocardiogram should delineate these features within the aortic arch and pulmonary artery (PA) anatomy. In selected cases, a computerized tomography (CT) scan may be performed to obtain accurate characterization of the PDA and branch PAs. It helps the operator to choose a vascular access site and to exclude unsuitable anatomies for PDA stenting, such as highly tortuous or long PDA and complex PA anatomy. Even when PDA stenting was contraindicated in cases of branch PA stenosis, some authors have implanted a stent in the PDA using techniques of intentional jailing of a branch PA and dilation/stenting through stent side cells [14,20]. It is recommended to choose the vascular access site in order to negotiate the PDA with the straightest trajectory which allows for easier maneuverability of guidewires/stent systems. Thus, a retrograde transfemoral artery approach can be chosen when the PDA arises from the proximal descending aorta with a straight and short course. In cases where the PDA originates from the underside of the aortic arch (vertical PDA), a percutaneous axillary artery or common carotid artery access may facilitate a straight access to the PDA and reduce procedural time [21,22]. Axillary artery access is a feasible and safe alternative, especially when performed with ultrasound guidance to ensure an isolated anterior wall puncture. An axillary artery is not an end artery and thus, when cannulated, arm perfusion is still guaranteed by the second intercostal and acromial artery [23]. In the case of a ventricular septal defect, an antegrade femoral or umbilical venous approach may be used for vertical PDAs, but PDA negotiation may still result, which would be cumbersome.

Most patients with duct-dependent CHD rely on prostaglandin E1 (PGE-1) infusion to maintain ductus patency. PGE-1 infusion discontinuation a few hours before the procedure produces a desired ductal constriction, providing an adequate landing zone for proper stent implantation [24]. During the procedure, PGE-1 should be kept in the line. The infusion can then be promptly restarted if there is a declining trend in saturations or in case of ductal spasm during wire or catheter advancement in the PDA (unless a stent is promptly implanted). A floppy 0.014″ guidewire is used to cross the PDA. For highly tortuous PDAs, the use of microcatheters will help to advance the wire through the turns. The stents typically used for PDA stenting are newer generation, flexible coronary artery stents, pre-mounted on low-profile balloons, available in either over-the-wire or monorail systems [12,25].

Though primarily bare metal stents (BMS) have been used, recently, drug eluting stents (DES) have emerged, as they bear a lower the risk of neointimal hyperplasia, thrombosis and restenosis [12]. Even though pharmacokinetic data in neonates suggest significantly lower clearance of sirolimus in neonates, no adverse clinical outcomes due to the prolonged immunosuppressive sirolimus levels was observed [12,26]. In a retrospective study of 71 infants who underwent ductal stenting for duct-dependent PBF, the luminal loss and unplanned reinterventions for desaturation were lower with DES than with BMS [27].

Stent diameter should be chosen based on the patient’s body size. Most authors agree to choose for deployment a 3.5–4 mm diameter stent for infants weighing >3.0 kg, a 3.5 mm diameter for those weighing 2.0–3.0 kg, and a 3-mm stent for those <2 kg [12,24,25,28,29]. Regarding the stent length, it is recommended to stent the entire PDA length to prevent early restenosis by ductal tissue constriction. While the choice of stent length is reasonably easy for straight and short PDAs, in cases of tortuous PDAs, measurements should be reassessed after wire crossing, as the PDA straightens. It is advisable to choose a stent slightly longer than the measured PDA length [25]. When the PDA is too long, it should be preferred to implant two short stents in a telescope technique and to start from the pulmonary end [30,31]. After stent deployment, a repeat angiography with the wire still in place is recommended to determine whether another stent should be implanted.

Regarding antiplatelet therapy, a low dose of aspirin at 5/mg/kg/day is recommended, for as long as stent patency is needed. The role of a dual antiplatelet therapy with the addition of clopidogrel is still unclear, but it should be considered, especially if a DES is used [25].

Major acute complications can include ductal spasm, which is uncommon (less than 1%), but can happen during guidewire manipulation across the PDA, and ductus arteriosus perforation or dissection. If the wire is in place, they can be addressed by implanting a stent quickly [24,28,31]. In a small number of cases, stent migration or malposition can also occur. The risk of this consequence is decreased by keeping the patient off PGE1, which promotes PDA constriction. Acute stent thrombosis is a relatively rare but life-threatening complication which can occur immediately after stent implantation or within few hours. If it occurs acutely during the procedure, it may be addressed with balloon angioplasty at the site of the thrombus. It may be necessary to maintain adequate ACTs for a period of time after the procedure [12], together with heparin administration, keeping in mind that its action is dependent on anti-thrombin III, usually deficient in neonates.

## 4. Pulmonary Valve Stenosis and Pulmonary Atresia with Intact Ventricular Septum

The gold standard of therapy for newborns with severe or critical valvular pulmonary stenosis is trans-catheter balloon valvuloplasty. Kan et al. [32] first described percutaneous balloon dilation for congenital pulmonary valve stenosis in 1982. Indications and timing of the procedure are based on PDA dependency and peak trans-valvular instantaneous gradient at echocardiography. Percutaneous pulmonary valvuloplasty is a well-standardized procedure. It requires a femoral, or sometimes umbilical, venous access. After a right ventricular angiography, with the aid of a floating end-hole catheter or a preformed end-hole catheter, such as a Bentson or a Judkins right curve, the valve is crossed with a floppy 0.014″ guidewire and then advanced across the PDA or the left pulmonary artery. The dilation should be performed with a 20 mm-long balloon with a diameter of 120–130% of the measured pulmonary valve annulus [33]. Infundibular spasm is frequent after balloon dilation and may require few hours of beta-blocker intravenous infusion. Infundibular spasm is well-tolerated in patients with PDA and PGE-1 infusion [34].

A thicker, dysplastic valve, concurrent supravalvular stenosis, and genetic abnormalities such as Noonan syndrome are linked to procedural failure, which is defined as residual gradient or right ventricular hypertension. In 5–20% of patients, recurrent valve stenosis requires repeat dilation [33,34].

In patients with pulmonary atresia and intact ventricular septum (PA/IVS), intervention is almost always required in the neonatal period [35]. It depends on the tricuspid valve annular size, the right ventricle dimensions and the presence of coronary artery abnormalities. In the setting of severe RV hypoplasia and/or RV-dependent coronary circulation, RV decompression is contraindicated, and the patient needs only a stable source of PBF, which can be achieved by PDA stenting. In case of restrictive atrial communication, a transcatheter atrio-septostomy may be also performed [36]. When technically feasible and not contraindicated, transcatheter perforation of the pulmonary valve should considered in case of membranous atresia, to establish an antegrade source of PBF, to offload the RV pressure and foster the RV growth.

Transcatheter perforation of the atretic pulmonary valve is usually achieved with the use of a radiofrequency coaxial system, which is anterogradely advanced from the RV [37]. More recently, chronic total occlusion (CTO) guidewires have been recently reported as an alternative to radiofrequency, with encouraging results [38,39,40]. Using a stiff end of a coronary wire may be a sustainable alternative to achieve pulmonary valve perforation in developing countries [41]. The procedural risk is not negligible and is associated with the possible RV or main PA perforation. According to a multicenter study from the Congenital Catheterization Research Collaborative, the risk of cardiac perforation is 10% [42]. According to a cohort study from the Pediatric Cardiac Care Consortium, mortality rates were higher when RV decompression was performed surgically (19%) rather than with a transcatheter approach (8%) [43].

Some patients may need a second source of PBF that can be achieved by PDA stent implantation during the same procedure or with a staged approach.

Re-interventions may be needed as part of the staging treatment because of suboptimal result of the first procedure (inadequate RV decompression), recurrence of RV outflow (RVOT) obstruction, or worsening of other lesions, such as tricuspid regurgitation. They can include repeat valvuloplasty, surgical augmentation of the RVOT, and creation of a second source of PBF [42,44].

## 5. Early Symptomatic Neonate with Tetralogy of Fallot

Primary repair of the majority of patients with tetralogy of Fallot (TOF) and good-sized confluent branch PAs is generally performed electively in the first year of life and has had excellent results. When the surgery is performed beyond the neonatal period, survival rate and post-operative course are better [45]. Severe RVOT obstruction and diminutive/hypoplastic branch PAs (eventually associated with multiple aortopulmonary collateral arteries) may require early intervention for severe cyanosis and duct-dependency. Coexisting comorbidities, such as prematurity, low body weight, younger age (<3 months), hypoplastic pulmonary arteries, and other conditions requiring noncardiac surgery, are associated with worse outcomes after complete primary surgical repair [46,47]. Moreover, TOF patients with severely hypoplastic central pulmonary arteries (Nakata index < 100 mm^2^/m^2^) are at risk of supra-systemic RV pressures and RV failure after primary repair [48]. In these patients, initial management is controversial and depends upon individual clinical basis and institution preferences. Some centers perform early repair, while others choose palliation such as a systemic-to-pulmonary surgical shunt, PDA stenting, or surgical augmentation of the RVOT. After the first description by Gibbs et al. [49], RVOT stenting has emerged as valuable short-term intervention used to secure forward flow in severe cyanotic TOF neonates with RVOT obstruction, diminutive branch PAs and high surgical risk.

The pulmonary valve is generally crossed with a 0.014″ supportive wire, whose tip is placed in either a distal pulmonary artery or in the descending aorta through the ductus arteriosus [50]. Low-profile, flexible, pre-mounted coronary stents significantly simplified the procedure, as they can be deployed without a long sheath. Stent diameter is chosen to be 1–2 mm larger than the minimum diameter of the infundibulum during diastole to ensure stent stability [50]. The stent should be long enough to cover the entire length of the RVOT.

In case of TOF and membranous pulmonary atresia, the pulmonary valve can be perforated (using radiofrequency or CTO wires) and then balloon-dilated as previously described, allowing the RVOT stent implantation thereafter [47].

The RVOT stent implantation is associated with improved growth and symmetry of branch PAs. Infants who undergo initial RVOT stenting have better survival when compared with aorto-pulmonary shunt [46,47,51], while clinical outcomes are similar when compared with early primary surgical repair [46,47].

Thus, stenting the RVOT can provide excellent palliation to allow for later elective primary repair. Anyway, RVOT stenting should be avoided in neonates who could benefit from potential valve-sparing surgery, since pulmonary valve annulus is part of the stent landing zone [47]. Currently, at some centers, surgical shunts and ductal stents are reserved only for infants in whom RVOT patency cannot be established (e.g., pulmonary muscular atresia), with unsuccessful RVOT intervention, or those with nonconfluent central pulmonary arteries [46,47,48].

Stent stenosis from neointimal proliferation, stent fracture, stent migration and size discrepancies secondary to somatic growth are factors limiting the durability of RVOT stents [52].

There are some potential issues related to the presence of a stent in the RVOT at the time of anatomic repair [53]. As previously said, it prevents any consideration for a valve-sparing anatomic repair, although RVOT stent implantation is generally performed in valves too small to be considered for conservative surgery. Second, the stent in the RVOT may complicate the repair, as it has been associated with significantly longer cardiopulmonary bypass time, with a rate of incomplete stent removal ranging from 5% [46] to 56% [54]. The additional time may be related to uneasy stent removal [54]. Other issues of partial stent removal include difficulties sewing the ventricular septal defect patch and impairment with aortic valve function [53].

## 6. Aortic Valve Stenosis

Neonatal aortic valve stenosis is a left ventricular outflow obstruction at valvular level, which presents and often requires treatment in the first month of life [55]. Invasive peak systolic gradients higher than 50 mmHg are associated with an increased risk of ventricular arrhythmias and sudden death; hence, urgent treatment is needed even in the absence of clinical symptoms [56]. In cases of left heart hypoplasia or multiple associated obstructive lesions, not all patients with aortic valve stenosis are good candidates for biventricular circulation, and a stage I Norwood procedure should be considered [57].

For patients with isolated aortic valve stenosis, the aim of initial treatment is to significantly relieve the left ventricular obstruction while minimizing regurgitation. In this perspective, balloon aortic valvuloplasty (BAV) and surgical aortic valvotomy are palliative procedures aiming to delay the aortic valve replacement. They both showed similar gradient reduction, occurrence of significant aortic regurgitation, long-term aortic valve replacement and improved survival rates. Due to its less invasive nature and faster recovery time, BAV is preferred by most centers over surgery [58,59].

The aortic valve can be crossed either with a retrograde (more frequent) or an antegrade approach (Figure 2). In the retrograde approach, access is obtained either via femoral, carotid, axillary or umbilical artery. With the aid of a shaped end-hole catheter, a floppy 0.014″ guidewire is looped in the left ventricular apex. The diameter of the selected balloon catheter should have a balloon-to-annular ratio of 0.9–1.0 and a length of 20 mm. The antegrade approach from the femoral or umbilical vein requires a transvenous entry in the left ventricle and aorta from the right atrium through an atrial communication. It is generally chosen in cases of a dilated and poor functioning left ventricle, which may permit easy manipulation of catheter and wires in its cavity [55].

The procedure has been shown to avoid or delay aortic valve surgery in long-term follow-up studies [60,61,62,63]. The challenge of BAV lies in reducing the left ventricular obstruction while limiting the amount of aortic regurgitation, aiming to delay the aortic valve replacement. In a large multicenter retrospective study from the IMPACT registry [64], BAV was successful in 70% of the cases. Most unsuccessful procedures were related to significant or worsening aortic regurgitation. Overall, there is a high incidence of adverse events (16%), which is greater in patients with critical aortic stenosis (30%) [64]. A more-than-mild aortic insufficiency occurs acutely in about 15% of cases [65].

Minimizing aortic regurgitation is therefore crucial. Balloon longitudinal displacement is a cause of valvular injury and insufficiency, and it is secondary to low wire stability, cardiac contractions and pulsatile flow. Several techniques have been used to stabilize the balloon during BAV [65,66,67].

Adenosine has been used to create a brief pharmacological cardiac standstill. Although the method was regarded as effective in the first experiences, asystole duration was neither predictable nor controllable. Moreover, adenosine does not prevent ventricular ectopies, which may occur spontaneously or as triggered by the guidewire or the balloon during inflation [65,66]. Rapid RV pacing is now the most common technique for balloon stabilization during BAV in older children and adults. It reduces stroke volume, blood pressure, and transvalvular flow, mimicking a cardiac standstill. Despite the evidence of its feasibility and efficacy, its use in neonates is still not widespread globally. It is associated with certain drawbacks in this population, such as prolonged operative times, increased risk of cardiovascular damage, additional venous access and increased risk of ventricular fibrillation [67].

The authors of the present work have recently applied rapid transesophageal atrial pacing to achieve balloon stabilization during neonatal BAV (Figure 2). In the first series of seven cases, it was safe and always allowed a significant relief of left ventricular obstruction while minimizing aortic regurgitation. Compared to right ventricular pacing, it has some advantages: it does not require additional vascular access and it is not at risk of cardiac or vascular perforation and ventricular arrhythmias.

## 7. Aortic Coarctation

In neonates and infants, aortic coarctation (CoA) has the common clinical pattern of congestive heart failure and has a poor prognosis if left untreated. Surgery is the first-choice treatment for native CoA in the first year of life, but in low birth-weight neonates, it is associated with increased mortality rates and morbidity rates, as well as recurrent CoA [68,69]. Low birth weight and prematurity are well-acknowledged risk factors for mortality in infants with congenital heart disease [69].

In selected cases, balloon angioplasty may be performed as a palliative strategy to stabilize neonates unable to undergo urgent surgical treatment due to severe clinical impairment [5,70]. Although the percutaneous treatment of aortic coarctation in neonates and infants remains controversial due to the occurrence of residual or recurrent stenosis and aneurysm formation at the dilation site [71], urgent balloon dilation can diminish mortality rates, providing a bridge to surgery for severely ill patients [23,72].

The procedure is typically performed via femoral artery access, but in the case of low-body-weight children, alternatives are the carotid or axillary artery access (Figure 3), which have an antegrade and straight trajectory to the aortic isthmus. The axillary artery access presents some technical advantages (Figure 3). It is easier to feel in smaller patients, especially in premature neonates and, in particular, in the presence of critical aortic coarctation, when femoral pulses are not palpable. Moreover, the axillary route proves critical in the case of concomitant low cardiac output due to failing left ventricle, which could make finding a femoral pulse even harder. It is not an end artery, and it does not need a surgical cutdown and repair, unlike the carotid artery access [23].

The initial balloon diameter is chosen to be two- or three-fold the minimal CoA diameter and not greater than the diameter of the aorta at the diaphragm. Generally, two or three dilations are employed, with a short inflation time (<10 s).

Balloon angioplasty for native CoA is highly successful acutely in infants less than 3 months of age, with a restenosis rate of approximately 50%, and with a lower rate of reintervention in discrete obstruction without aortic arc hypoplasia [73]. In a recent retrospective study of 68 patients with native aortic coarctation, Sandoval et al. [74] showed that balloon angioplasty was effective and safe in infants aged 3 to 12 months, with outcomes comparable to those in older children and adults. Repeat angioplasty or stenting can avoid the need for surgery in most patients.

Considering that smaller and younger patients have a greater rate of restenosis, stent implantation is another palliative measure for preterm neonates with CoA. After the first report in 2002 by Radtke et al., with the use of rigid bare metal stent [75], CoA stenting using pre-mounted coronary stents became a safe alternative to treat the acute symptoms and to bridge the patients to surgery [76,77]. The surgical repair, including stent removal, can then be performed once the patient has reached a sufficient body weight. However, stents can cause significant trauma and tissue loss during surgical repair [78]. Bioresorbable stents offer a theoretically attractive alternative with less tissue damage during surgical removal. Although the first report of a magnesium-based scaffold in a less-than-2 kg preterm neonate with CoA allowed a short-term bridging palliation to surgery [79], the authors observed early restenosis and stent failure due to loss of radial force of the scaffold. Therefore, present bioresorbable stents may have limited applicability in very-low body weight patients with aortic coarctation requiring a long-term bridge-to-surgery.

## 8. Hybrid Stage I Palliation

Neonates with HLHS and its variants have diminutive left-sided structures and systemic outflow obstruction. Subsequent systemic, cerebral, and coronary circulations are dependent on ductus arteriosus patency [80]. The hybrid stage I palliation has been developed as an alternative to the Norwood procedure for these patients. The aim is to defer the more complex surgical treatment and associated cardiopulmonary bypass until later in infancy and to perform a lesser-risk first stage procedure using a combination of transcatheter and surgical techniques. The early results of the hybrid approach, according to Akintuerk et al. [81] and Galantowicz et al. [82], have encouraged the application of this strategy to reduce the deleterious impact of the conventional surgery on higher-risk infants. Risk factors include low body weight, extracardiac abnormalities, severe atrioventricular valve regurgitation, and ventricular dysfunction. The technique and patient selection vary greatly from center to center, but the hallmarks of the hybrid procedure include stent implantation in the ductus arteriosus, PAs flow restriction and creation of a stable non-restrictive atrial communication.

The procedure is carried out during PGE-1 infusion. After performing the branch pulmonary artery banding via median sternotomy, the stent implantation of the ductus arteriosus is carried out with surgical placement of a delivery sheath in the main pulmonary artery (single-step true hybrid procedure [82], Figure 4) or through a percutaneous approach (the Giessen technique). Ductal stenting is generally performed after PAs banding, because there is a theoretical risk of stent migration during intraoperative manipulation [83]. A completely transcatheter palliation was recently successfully performed by Schranz et al. [84]. They achieved pulmonary artery banding by implanting bilaterally a vascular plug manually modified a pulmonary flow restrictor.

The stent diameter should exceed the minimal diameter of the ductus by at least 1 mm or 2 mm. Thus, sometimes it may be larger than the descending aorta diameter, varying from 7 to 10 mm. The entire length of the ductus into the aortic stump should be entirely covered to avoid retrograde aortic arc obstruction. Either self-expandable or balloon-expandable stents can be used. Balloon-expandable stents have a higher radial force, which is useful when asymmetrical duct constriction is observed [31]. In case of percutaneous approach to ductal stenting, self-expandable stents have some advantages, as they are specifically designed for the HLHS duct, have a lower profile and can conform to ductal anatomy [85].

In case of restrictive atrial communication (quite frequent in HLHS patients), a balloon atrial septostomy or atrial septal stenting is performed (Figure 1).

The risk of retrograde aortic arch obstruction is potentially life-threating, because it leads to coronary hypoperfusion and cerebral ischemia. It may occur early, shortly after the procedure, because of PDA constriction (if not fully covered), or later, due to in-stent stenosis and intimal proliferation [86]. Reinterventions due to a retrograde aortic arch obstruction are required in about 20–30% of patients, and are usually managed with balloon dilation or ductal re-stenting.

For centers practicing the hybrid stage I palliation as a standard of care, short-term, interstage and mid-term mortality rates are equivalent to multicenter outcomes following traditional Norwood procedure [87,88]. Favorable neurodevelopmental data are reported, since cardiopulmonary bypass and deep hypothermic circulatory arrest are avoided [89]. A significant selection bias affects these results due to institutional differences in patient selection. Major concerns associated with the hybrid stage I procedure include the unrepaired hypoplastic aortic arch, with the associated risk of cerebral and coronary hypoperfusion due to retrograde aortic arch obstruction, mechanical distortion of the branch PAs and its impact on future Fontan circulation, and significantly greater surgical complexity during comprehensive stage II palliation.

## 9. Patent Ductus Arteriosus Closure in Low-Body Weight and Preterm Infants

Patent ductus arteriosus (PDA) is highly prevalent in premature infants, with an incidence at 40 days of life >50% in extremely low-birth weight neonates [90]. PDA is associated with an eightfold increase in mortality rates and with multiple comorbidities such as necrotizing enterocolitis, chronic lung disease and intraventricular hemorrhage [91]. Hemodynamic significance of the PDA is established using clinical and echocardiographic criteria. Although successful in nearly all cases, surgical PDA ligation through a limited left thoracotomy has been associated with procedural complications, including pneumothorax, bleeding, phrenic nerve palsy, wound infection, vocal cord paralysis and chylothorax. Pharmacological PDA closure is only partially successful, and may be associated with significant side-effects.

Transcatheter PDA closure in patients weighing >5 kg is the first-choice treatment, as it is a safe and effective procedure [89]. In the past 15 years, transcatheter PDA closure has been extended to treat smaller patients, with reports of procedures in neonates smaller than 1000 g. Zahn et al. [92] and Sathanandam et al. [93] demonstrated successful PDA closure in about 90% of patients using two different devices, with no severe adverse events. The encouraging outcomes of transcatheter PDA occlusion in preterm newborns have led to the development of novel devices and an increase in practice. In a multicenter, prospective, non-randomized trial of 200 neonates weighing ≥700 g using the Amplatzer Piccolo Occluder (Abbott), the implant success rate was 95.5% overall and the rate of major complication was 2%. In the cohort of patients < 2 kg, the success rate was 99% [94]. Most observed complications were device embolization, left PA stenosis, aortic coarctation and tricuspid valve damage. In order to avoid acute arterial injury and limb loss, the procedure is now carried out exclusively via femoral venous access. To minimize the contrast dye administered, only one aortogram is performed once the catheter is placed in the descending aorta from the main PA through the PDA. Transthoracic echocardiographic guidance is used to exclude the presence of obstruction on the aortic or pulmonary side after device placement and before its release (Figure 5).

Compared to SLP, transcatheter PDA closure shows faster weaning of respiratory support [95] and a lower incidence of post-ligation syndrome [96]. Optimal timing for PDA closure is a matter of debate. However, as experience grows, some institutions support early intervention in order to prevent the negative consequences of extended mechanical ventilation and supplementary oxygen administration. In a retrospective study of 80 infants born <27 weeks, weighing <1 kg at birth and <2 kg at transcatheter PDA closure, patients who had PDA closure after 8 weeks of life had elevated PA pressure. It may be beneficial to close the PDA in the first 4 weeks of life and before the onset of elevated PA pressure.

## 10. Diagnostic and Prognostic Yield of Cardiac Catheterization

A diagnostic phase usually precedes neonatal transcatheter interventions and provides important prognostic information [5]. RV angiography performed before pulmonary valvuloplasty informs about long-term outcomes based on pulmonary annulus size and valve morphology [97]. Diagnostic phase is essential in some situations. For example, in PA-IVS it is required to exclude a RV-dependent coronary circulation, allowing clinicians to decide whether to decompress the right ventricle and establish RV-to-PA continuity [36]. In patients with tetralogy of Fallot and pulmonary atresia, catheterization can provide details of aortopulmonary collateral supply to the lungs and PA size, distinguishing a double supply from a single supply for each pulmonary segment [98]. Hemodynamic and angiographic data collected at the end of the intervention are sources of prognostic information. For example, final pressure gradients at the end of aortic and pulmonary valvuloplasty are independent indicators of reoperation-free survival [97,99,100].

## 11. Conclusions

Many transcatheter procedures are now among the first-choice treatments for newborns with critical CHD. Some of these interventions can be considered curative, while some others may only offer short-term palliation prior to surgical repair [101]. Accumulating evidence supports percutaneous interventions for PDA closure for aortic coarctation, RVOT obstructions and duct-dependent PBF as alternatives to conventional management strategies. The use of lower-profile catheters, balloons and stents, along with innovative vascular access approaches, has made it possible to treat percutaneously the low body-weight neonates, who have a higher surgical risk. Although cardiac catheterization is associated with complications, the key elements for success are accurate planning, use of proper hardware (e.g., microcatheters and dedicated wires), body temperature control, and procedural time optimization. Ongoing and planned multicenter studies will inform optimal management strategies for neonates with critical CHD.

## Figures and Tables

**Figure 1 diagnostics-13-02673-f001:**
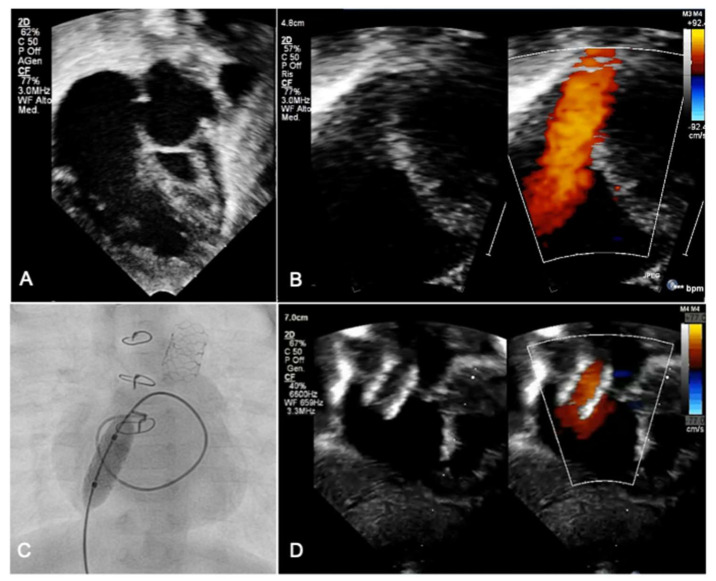
In an infant with HLHS (**A**), atrial septal communication became restrictive (**B**) several days after stage I hybrid procedure. Serial static balloon septoplasties were performed before atrial septal stent implantation (**C**). At the echocardiogram, flow across the atrial stent was adequate to offload the left atrium pressure (**D**).

**Figure 2 diagnostics-13-02673-f002:**
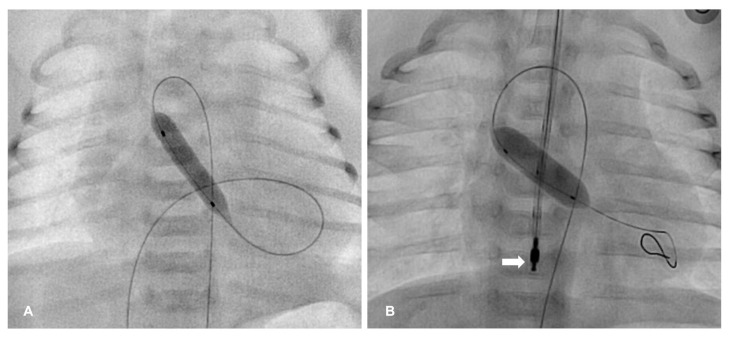
Balloon aortic valvuloplasty. (**A**) The valvuloplasty is performed with an anterograde approach in an infant with severe aortic valve stenosis and left ventricular dilation and dysfunction. (**B**) In the retrograde approach, wire stability is achieved with rapid atrial pacing using a transesophageal electrode (arrow).

**Figure 3 diagnostics-13-02673-f003:**
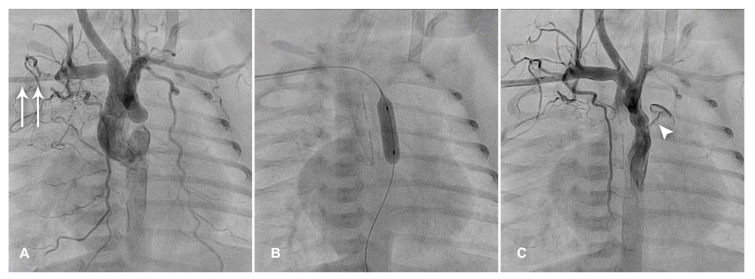
(**A**) Right axillary artery was cannulated with a 4 Fr sheath (arrows). The aortogram shows a severe aortic coarctation. (**B**) Balloon angioplasty is performed. (**C**) The repeat angiography documented a good procedural result, with the patency of a tiny ductus arteriosus (arrowhead).

**Figure 4 diagnostics-13-02673-f004:**
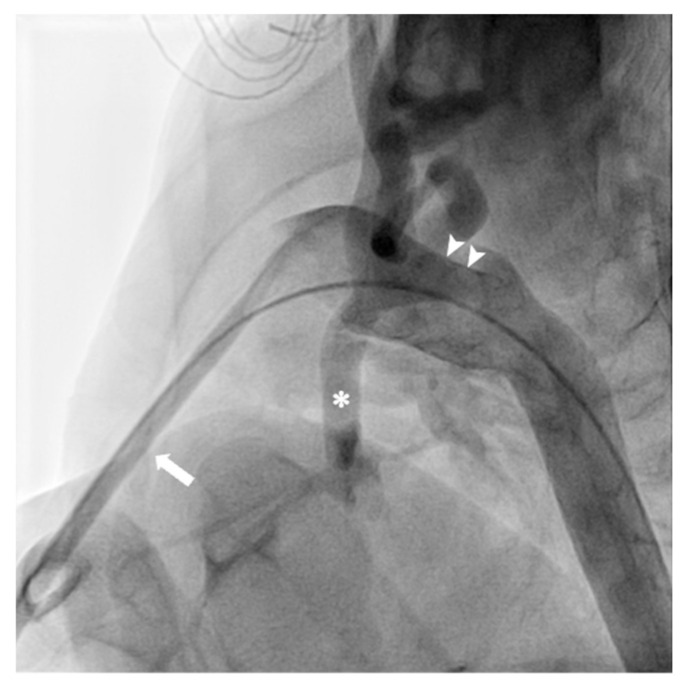
After performing the branch pulmonary artery banding via median sternotomy, the stent implantation of the ductus arteriosus (arrow-heads) is carried out with surgical placement of a delivery sheath (arrow) in the main pulmonary artery. Adequate retrograde flow to hypoplastic ascending aorta (asterisk) and coronary circulation is documented.

**Figure 5 diagnostics-13-02673-f005:**
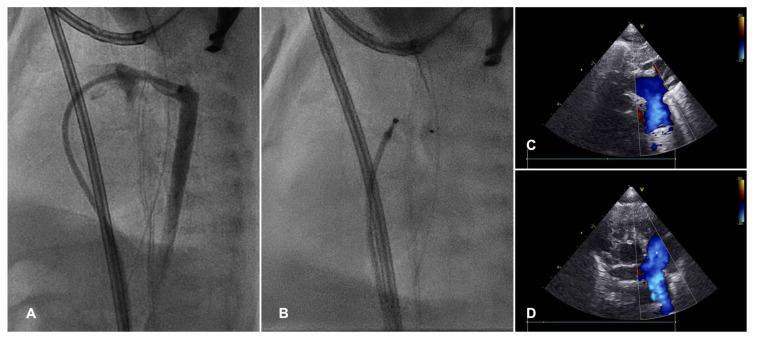
Transcatheter patent ductus arteriosus closure in a premature infant. (**A**) Angiographic PDA assessment with injection through the delivery sheath anterogradely positioned through the PDA into the proximal descending aorta. (**B**) The device is positioned within the PDA length. Echocardiographic imaging confirms correct device position prior to detachment, excluding residual shunt, aortic arch obstruction (**C**) or left pulmonary artery stenosis (**D**). Courtesy of Biagio Castaldi, MD, Pediatric Cardiology, Department of Women’s and Children’s Health, University of Padua, Padova, Italy.

**Table 1 diagnostics-13-02673-t001:** Main indications and possible complications in main neonatal transcatheter procedures [5].

Transcatheter Procedure	Indications	Complications
Balloon atrial septostomy	Restrictive atrial communication in: D-TGA, DORV with TGA physiology (parallel circulations), tricuspid atresia, PA-IVS with RVDCC or hypoplastic RV (right heart obstructive CHD).	Embolization, IVC laceration, cardiac perforation.
Atrial septum perforation/static balloon septoplasty and/or stent implantation	HLHS complex with restrictive atrial communication	Stent embolization, cardiac perforation. Late: in-stent stenosis.
PDA stenting	Ductus-dependent PBF until next stage of surgery (repair or II stage SV palliation); PA-IVS after PV perforation (inadequate anterograde PBF).	Acute: stent thrombosis, migration and embolization. Late: in-stent stenosis, PA branch jailing.
Pulmonary valvuloplasty	Critical PVS, PVS and RV dysfunction, severe PVS (RV-PA invasive peak gradient ≥ 40 mmHg)	Tricuspid valve injury, cardiac perforation.
Pulmonary valve perforation	PA-IVS with membranous PA (excluded RVDCC); ToF-PA prior to RVOT stenting.	Tricuspid valve injury, cardiac and pulmonary artery perforation.
RVOT stenting	Severe RVOT obstruction in ToF and DORV with ToF physiology with diminutive/hypoplastic PAs and surgical risk factors: prematurity, low BW, age <3 months, extracardiac condition requiring surgery	Stent migration, in-stent stenosis, stent fracture, obstruction due to somatic growth.
Aortic valvuloplasty	Critical AVS, AVS and LV systolic disfunction, severe AVS (LV-Ao invasive peak gradient ≥ 50 mmHg)	Significant aortic regurgitation, mitral valve injury, cardiac perforation.
BAA/stent implantation	CoAo and concomitant high surgical risk: prematurity, low BW, LV systolic dysfunction, LCO state	BAA: Acute aortic dissection, aortic aneurysm. Stent: in-stent stenosis, obstruction due to somatic growth.
Hybrid stage I palliation	HLHS complex (institutional preference, high surgical risk)	Stent migration, retrograde/antegrade aortic obstruction, inadequate PBF restriction.
Transcatheter PDA closure	Hemodynamic significant PDA (clinical and echo indices) after failed medical therapy or NSAID contraindications	LPA stenosis, CoAo, device migration/embolization, tricuspid valve injury, IVC laceration, residual shunt/hemolysis.

AVS: aortic valve stenosis; BAA: balloon aortic angioplasty; BW: body weight; D-TGA: D-transposition of great arteries; DORV: double-outlet right ventricle; HLHS: hypoplastic left heart syndrome; IVC: inferior vena cava; PA: pulmonary atresia; PA-IVS: pulmonary atresia with intact ventricular septum; PBF: pulmonary blood flow; PDA: patent ductus arteriosus; PVS: pulmonary valve stenosis; RVDCC: right-ventricle-dependent coronary circulation; RVOT: right ventricular outflow tract; SV: single ventricle; TGA: transposition of great arteries; ToF: tetralogy of Fallot; ToF-PA: tetralogy of Fallot with pulmonary atresia.

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
