# Peer review of "Transcatheter Interventions for Neonates with Congenital Heart Disease: A Review"

_diagnostics, 2023, doi:10.3390/diagnostics13162673_

Round 1
Reviewer 1 Report
This review is an interesting summarization of the current knowledge on the frontier therapeutic applications of transcatheter intervention operations on congenital heart diseases. The authors did a great job on archiving these information according to congenital heart diseases (CHD), including ductus-dependent pulmonary blood flow CHD, pulmonary valve stenosis/pulmonary atresia, tetralogy of Fallot, and aortic valve stenosis. I appreciate that the authors also covered the assistant medicine therapies and possible adverse events after these interventions. Overall, the manuscript and figures are clearly organized, and provide an updated information. I listed several concerns need to be addressed.
1. A table is needed to recap all the transcatheter intervention techniques mentioned, together with the indications and adverse effects of this technique. For example, under the category of balloon atrial septostomy, tricuspid atresia and pulmonary atresia are the indications and embolization is the adverse effect.
2. About the figures, are these images from cited papers or the authors’ own collection? If they are from cited papers, please state the citation in the legend and get the citation agreement from the original authors. If they are from your own collection, please support information consents from patients.
3. In adults, transcatheter intervention is also useful techniques for disease diagnosis, classification and severity evaluation (PMID: 23906495, 25676815). For the purpose of novelty, could you also talk about any application of the intervention techniques on diagnosis, evaluation, re-evaluation after intervention and prognosis prediction of CHD?
Author Response
The authors thank the reviewer for his/her valuable comments and precious suggestions, which will improve the overall quality of the work.
Point 1. We totally agree with the reviewer, as a summary table was certainly needed. We added the suggested table (page 2).
Point 2. The images are from the authors’ own collection with the exception of Figure 5. In the “Images” paragraph at the end of the manuscript (page 13), we added the patients’ family member informed consent to image publication and Dr. Castaldi’s permission to Figure 5 submission.
Point 3. As the reviewer pointed out, cardiac catheterization carries important diagnostic and prognostic data. We added a final section “Diagnostic and prognostic yield of cardiac catheterization”.
Reviewer 2 Report
This article systematically introduces the application of transcatheter intervention in various common types of congenital heart diseases, and is a good guide introduction for beginners. As a state-of-the-art review, it might be more valuable to introduce the latest developments and research reports of transcatheter intervention. For example, what improvements in hospital length of stay, mortality, and cost for atrial septal communication when compared transcatheter intervention with traditional surgical approaches, and what are the current limitations of transcatheter intervention when applied in each type of congenital heart disease? Is there any room for improvement for transcatheter intervention?
Minor:
Line 297, The period ”.” after regurgitation is missing.
Line 464-465, please clearly demonstrate where the images in Fig.5 were adopted from
Author Response
The authors thank the reviewer for his/her valuable comments and precious suggestions, which will improve the overall quality of the work.
Major.
1) A comparison between catheter-based intervention and standard/surgical approach was implemented in section 2 and 4 and is now present in all sections.
2) Limitations and adverse events, which lacked in some sections, are now summarized in the added Table.
3)Areas of possible improvement are highlighted whenever possible: e.g. total transcatheter palliation for HLHS, bioresorbable stents in aortic coarctation, transesophageal atrial pacing in balloon aortic valvuloplasty.
Minor.
1) A full stop was added were indicated.
2) We added “Images” paragraph at the end of the manuscript (page 13), where we reported Dr. Castaldi’s permission to Figure 5 submission.
Reviewer 3 Report
Would you be able to add three-dimensional echocardiography images?
Especially having pre-process images would be easy to understand the cases.
Author Response
The authors thank the reviewer for his/her valuable suggestions.
Point 1. Transthoracic or transesophageal three-dimensional echocardiography (3D echo) images are not routinely acquired in neonates, because of technical limitations. Given the high heart rate, temporal and spatial resolution is too low for 3D echo in neonates, and generally bidimensional imaging is sufficient for diagnosis and procedural planning.
Point 2. We added the pre-procedural imaging in the Figure1 and added symbols to Figure 4, in order to make it easier to understand. Figure 3 and 5 already contained a pre-interventional angiogram. Figure 2 was intended to compare two different approaches to balloon aortic valvuloplasty and pre-dilation echocardiogram would appear pretty standard.
Reviewer 4 Report
This is a review article of transcatheter interventions for naonates with congenital heart disease.
The references are missing in many paragraphs, therefore the readers cannot tell which is the authors' original comments. Detailed instructions for procedures seemed to be those performed in authors' facility, however it is not clear if those are standard procedures in the world.
Indications and complications of each procedures are not clearly stated.
The review should be thoroughly revised to be organized.
English language is mostly understandable, however some expressions are not appropriate (for example: "By the way" in some paragraph). Misuse of terms are observed occasionally.
Author Response
The authors thank the reviewer for the comments, as the changes they elicited improved the quality of the work.
Point 1a. The authors carefully revised the text and added bibliographic references where missing.
Point 1b. There is a standard procedure for only few neonatal interventions. Even the most dated and shared ones have major and minor variations in each cath lab. Many of the interventions reported in the manuscript are not performed in all centres, depending on the preferences and expertise of individual institutions. Some procedures have represented a concrete and innovative alternative to traditional surgery. The authors decided to report procedural details of these interventions, supporting them with bibliographic references (further implemented after this revision). For the above reasons, we could not describe all procedural variants, but this was beyond the scope of the review. The selection of the described procedural variant was possibly biased by authors' practice.
Point 2. As requested, we added a table with indications and complications of each procedures.
Reviewer 5 Report
Here are my conclusion: Very interesting and complete review of how to deal with different neonatal heart diseases with innovative techniques such as transcatheter procedures. The various pathologies are examined, and the clinical results compared to traditional treatments are evaluated. Excellent iconography that makes both the technique and the result very understandable. You can proceed to publication. English OK Best regards
Author Response
The authors are grateful for the positive comments on the manuscript, which increase the possibility of its publication.
Round 2
Reviewer 1 Report
This review is an interesting summarization of the current knowledge on the frontier therapeutic applications of transcatheter intervention operations on congenital heart diseases. The authors did a great job on archiving the information on multiple congenital heart diseases (CHD), including ductus-dependent pulmonary blood flow CHD, pulmonary valve stenosis/pulmonary atresia, tetralogy of Fallot, and aortic valve stenosis. I appreciate that the authors also covered the assistant medicine therapies and possible adverse events after these interventions. Overall, the manuscript and figures are clearly organized, and provide an updated information. The authors responded well to my questions and made sufficient revisions. After explanations, the conclusion is less vulnerable.
Reviewer 2 Report
The authors have addressed my concerns. I have no further comments.
Reviewer 4 Report
The authors added references, but there are still some descriptions which requires reference.
The authors added "Table". Is this modified from reference 5? How about the copyright?
This article includes the procedures and techniques in the authors' facility. If this is a review article, the authors should clarify which is their original method and which is from reference.
If the authors use more subtitles such as "indication", "procedures", and "complications", the article will be easier to read.
There are still some typos and grammatical errors throughout the manuscript.